# Infectiousness of *Leishmania major* to *Phlebotomus papatasi:* differences between natural reservoir host *Meriones shawi* and laboratory model BALB/c mice

Barbora Vojtková[1], Tomáš Bečvář[1], Lenka Pacáková[1], Daniel Frynta[2], Nalia Mekarnia[3,4], Kamal Eddine Benallal[1,5], Petr Volf[1], Jovana Sádlová[1]*

1 Department of Parasitology, Charles University, Prague, Czech Republic, 2 Department of Zoology, Charles University, Prague, Czech Republic, 3 UR ESCAPE – USC Anses, Faculty of Pharmacy, Université de Reims Champagne-Ardenne, Reims, France, 4 UMR BioCIS, CNRS, Université Paris-Saclay, Orsay, France, 5 Arboviruses and Emergent viruses, Institut Pasteur d'Algérie, Algiers, Algeria

* sadlovaj@natur.cuni.cz

## Abstract

Host infectiousness to insect vectors is a crucial parameter for understanding the transmission dynamics of insect-borne infectious diseases such as leishmaniases. Despite their importance, critical factors influencing the outwards transmission of *Leishmania major,* including parasite distribution within the host body and the minimum number of skin amastigotes required for vector infection, remain poorly characterized. To address these gaps, we studied these parameters in the natural North African reservoir host *Meriones shawi* and in BALB/c mice infected with a low parasite dose. Using qPCR, we quantified *Leishmania* loads in different zones (regions) of infected ear pinnae, whereas microscale infectiousness was evaluated via microbiopsies and fluorescence microscopy. The amastigote distribution within infected ears was heterogeneous, with pronounced differences between the lesion center, lesion margin, and visually unaffected surrounding skin. *Phlebotomus papatasi* females that fed in areas where no amastigotes were detected via microscopy did not become infected. In *M. shawi*, lesion margins have emerged as the most effective source of infection. The number of amastigotes at bite sites where sand fly females became infected ranged from 4--500, with as few as 2--10 amastigotes sufficient to initiate vector infection. This low infection threshold was confirmed by experiments in which *P. papatasi* was fed through a chick-skin membrane. In contrast, the BALB/c mouse model showed only minor differences in infectiousness between lesion centers and margins. The minimum infectious dose in BALB/c mice was approximately 100 times greater than that in *M. shawi*, with successful infections occurring at sites containing 1,500–10,000 amastigotes. These findings advance our understanding of *Leishmania* transmission by addressing critical knowledge gaps and enabling more accurate modelling of cutaneous leishmaniasis

**Data availability statement:** Data are contained within the article or Supporting information.

**Funding:** This research was funded by the Czech Science Foundation (GACR, https://gacr.cz, project No. 23-06299S to PV, JS, TB, BV). The collaboration between Charles University and Institut Pasteur d'Algérie and establishment of M. shawi colony was funded by the European Union's Horizon 2020 RIIP-LeiSHield MATI-RISE research and innovation programme under the Marie Skłodowska Curie (https://marie-sklodowska-curie-actions.ec.europa.eu/, grant agreement No. 778298, to JS, NM, KEB). The funders had no role in study design, data collection and analysis, decision to publish, or preparation of the manuscript.

**Competing interests:** The authors have declared that no competing interests exist.

epidemiology. Moreover, this study highlights the importance of incorporating natural host models in research, as the dynamics of disease progression and transmission parameters can differ significantly between natural hosts and standard laboratory models.

## Author summary

*Leishmania major* is a unicellular parasite that causes cutaneous ulcerative disease and circulates between rodent reservoir hosts and the blood-feeding insects, phlebotomine sand flies. In this study, we investigated transmission parameters important for understanding and modelling the epidemiology of leishmaniasis using two model animals, the natural host *Meriones shawi* and the standard laboratory mouse strain BALB/c. We found that *Leishmania* parasites are unevenly distributed in the infected ear pinnae of rodents, with the highest numbers observed in the centres and margins of the lesions. We subsequently analysed individual bite sites via fluorescence microscopy of microbiopsies, which mimic the volume and depth of penetration of sand fly bites. For *M. shawi,* the lesion margins proved to be the most effective sources of sand fly infections, with the ingestion of as few (2--10) amastigotes sufficient to establish infection. On the other hand, in BALB/c mice, sand flies become infected when they feed at both centers and margins of the lesions, and the threshold is at least 1000 parasites. Thus, this study not only provides valuable data for modelling the *Leishmania* transmission cycle but also demonstrates the importance of model organism selection in the study of host–parasite and vector interactions.

## Introduction

Leishmaniases are neglected diseases caused by *Leishmania* spp. (Kinetoplastida: Trypanosomatidae), which have a digenetic life cycle involving blood-feeding insects and vertebrate hosts. In hosts, these diseases present three different clinical forms. The most severe infections are visceral leishmaniasis (VL), which affects visceral organs, and mucocutaneous leishmaniasis (MCL), which leads to destruction of the nasal and oral mucosa and cartilage. The most common form of the disease is cutaneous leishmaniasis (CL), characterized by a nonpainful ulcerative skin lesion at the site of the vector bite [1]. In 2022, over 200,000 new cases of CL were reported, mainly from the epidemiological "hotspot" region of the Eastern Mediterranean and Algeria, which accounted for 79% of global CL cases [2]. However, given that CL is a disease with relatively low morbidity, the official case numbers are likely underestimated. Detailed studies in endemic sites, such as the Mediterranean [3] or Central America [4], have revealed significantly higher case numbers than officially reported.

 Cutaneous leishmaniasis caused by *L. major* is widespread from North Africa to Central Asia. The main reservoir hosts (RH) are rodents from several genera,

including the fat sand rat *Psammomys obesus* and Shaw's jird *Meriones shawi* in North Africa and the great gerbil *Rhombomys opimus* in Central Asia [5]. These reservoir animals are the main source of infection for phlebotomine sand flies (Diptera: Psychodidae), which act as vectors, transmitting parasites to other vertebrate hosts, including humans, who are incidental hosts. *Phlebotomus papatasi* is the principal vector of *L. major* over a wide geographical range that extends from North Africa to India, with breeding sites in rodent burrows [6]. In this specific vector, parasite attachment to the midgut epithelium is controlled by the surface glycoconjugate lipophosphoglycan, which selectively binds to the midgut galectin receptor [7].

Two main factors determine whether a competent vector feeding on an infected host becomes infected: the homogeneity of parasite distribution in the skin and the number of ingested parasites [8]. For example, distribution heterogeneity of *L. donovani* in host tissues (patchiness at both the macro- and microscales) has been recently described in immunodeficient RAG-2 mice. The authors also suggested an infection threshold of 500–1000 amastigotes required for a success of natural infection. They applied a mathematical model showing that while patchy distribution of *Leishmania* reduces the expected number of infected sand flies, the clustering of parasites at specific bite sites increases the infectious load of sand flies feeding on these patches [9]. However, similar investigations using natural host models and cutaneous *Leishmania* species are lacking.

To address this gap, we studied *M. shawi* originating from an endemic site in Algeria and BALB/c mice, a commonly used laboratory model for CL, both infected with a low dose of *L. major.* In previous work, we demonstrated that even asymptomatic individuals of *M. shawi* can serve as effective sources of infection for sand fly vectors [10]. Here, we report the spatial distribution of parasites in infected ear pinnae at the microscale and the threshold number of *L. major* required for infection in the proven vector *Phlebotomus papatasi*.

## Methods

### Ethics statement

Animals were maintained and handled in the animal facility of Charles University in Prague following institutional guidelines and Czech legislation (Act No. 246/1992 and 359/2012 coll. on Protection of Animals against Cruelty in present statutes at large), which complies with all relevant European Union and international guidelines for experimental animals. All the experiments were approved by the Committee on the Ethics of Laboratory Experiments of Charles University in Prague and were performed with permission no. MSMT-31778/2019–6 of the Ministry of Education, Youth and Sports of the Czech Repubic. The investigators are certified for experimentation with animals by the Ministry of Agriculture of the Czech Republic.

### Sand flies, parasites and rodents

The colony of *P. papatasi* originating from Turkey was maintained in the insectary of the Department of Parasitology, Charles University in Prague, under standard conditions (26 °C on 50% sucrose, humidity in the insectary 60–70% and 14 h light/10 h dark photoperiod), as described previously [11].

The human isolate *L. major* MHOM/DZ/2009/LIPA100/MON-25 from the M'Sila region in Algeria was used. To evaluate the parasite distribution at the microscale, *Leishmania* were fluorescence labelled with m-Scarlet according to the methodology described by Dean et al. (2015) [12]. Promastigotes were cultured in M199 medium (Sigma–Aldrich, Merck, Darmstadt, Germany) containing 10% heat-inactivated foetal bovine calf serum (FBS; Gibco, Thermo Fisher Scientific, Waltham, MA, USA) supplemented with 1% BME vitamins (Basal Medium Eagle, Sigma–Aldrich, Merck, Darmstadt, Germany), 2% sterile human urine and 250 µg/mL amikacin (Amikin, Bristol-Myers Squibb, New York, NY, USA).

Two rodent species were used: BALB/c mice originating from AnLab s.r.o. (Prague, Czech Republic) and *Meriones shawi*. The breeding colony of *M. shawi* was established at the animal facility of the Pasteur Institute of Algiers from animals originating in M'Sila, Algeria, and then bred at the Animal Facility of the Faculty of Science, Charles University, in

Prague. BALB/c mice were kept in T2 and T3 breeding containers (Velaz s.r.o., Prague, Czech Republic), and *M. shawi* were kept in T4 containers (Velaz, Prague, Czech Republic). The containers were equipped with bedding (SubliCZ.cz, Sojovice, Czech Republic), nesting material (Woodwool, Miloslav Vlk s.r.o., Ratibor, Czech Republic), and hay (Krmne smesi Kvidera s.r.o., Nezvestice, Czech Republic). Animals were provided with the standard feed mixture Myška (SubliCZ.cz, Sojovice, Czech Republic) and water ad libitum, with a 12 h light/12 h dark photoperiod, temperature 22–25 °C and humidity 40–60%.

## Experimental infections of rodents

Rodents were anaesthetized with a mixture of 50 mg/kg ketamine and 5 mg/kg xylazine (*Meriones shawi*) or 62.5 mg/kg ketamine and 25 mg/kg xylazine (BALB/c mice) and infected with parasites derived from the thoracic midgut (TMG) of experimentally infected *P. papatasi*. The methodology of infection is described in detail in [10]. Briefly, parasites from log-phase cultures were washed in saline and resuspended in heat-inactivated defibrinated ram blood at a concentration of $2 \times 10^6$ promastigotes/mL. The female sand flies were allowed to feed the infectious suspension through a chick-skin membrane. Engorged females were maintained under standard conditions until day 8 PBM, when mature infections had developed. Sand fly guts were dissected, and parasites from the TMG were pooled in sterile saline and counted in a Bürker chamber. Dissected and lysed SGs were added to the suspension, and 5.5 µL of the mixture was intradermally injected into the left ear pinnae. The infective dose per rodent was 7,000 or 14,000 parasites, equivalent to 1 or 2 infected TMGs with the addition of 0.5 SGs. The animals were checked weekly for external signs of the disease, and their infectiousness to the sand flies was continuously tested. After careful euthanasia of the rodents by cervical dislocation under anaesthesia, the tissues were collected for molecular analysis. In *Meriones shawi,* the experiment was mostly terminated at week 30 p.i. (in 21 from 32 animals), but to cover a wider range of disease manifestations and to obtain samples from asymptomatic phases of infection, some animals were euthanized at different time intervals: from week 15 p.i., when lesions appeared, to week 38 p.i., when lesions were healed.

Since the high burden of *Leishmania* in the ear pinnae persisted in almost all the animals until the end of the experiment, we selected this tissue for a detailed analysis of parasite distribution and associated transmission parameters. The inoculated ears were removed, divided according to zones (centre of the lesion, margins of the lesion and intact skin), placed into plastic tubes filled with 100 µL of distilled water and stored at −20 °C for subsequent *Leishmania* DNA extraction.

In addition, we evaluated transmission parameters in BALB/c mice, the most commonly used laboratory model for cutaneous leishmaniasis research. These mice were infected and treated in the same manner as *M. shawi*, but the duration of the experiment was limited to 11 weeks because of the rapid onset and destructive nature of skin lesions in this susceptible mouse strain [13].

## Infectiousness of rodents at the microscale

The infectiousness of 20 *M. shawi* and 5 BALB/c mice to sand flies was tested at the microscale twice during the experiment: at weeks 15–16 p.i., when skin lesions started to appear and weeks 25–30 p.i., when lesions were fully developed. Infected rodents were anaesthetized with ketamine/xylazine, and their left ear (inoculated) was exposed to sand fly bites. Five-day-old female *P. papatasi* were individually placed in small plastic tubes covered with fine nylon mesh and allowed to bite successively. The bite sites were marked immediately (Fig 1A), and a microbiopsy (MB) was taken from each bite site. Harper devices (Trajan Scientific and Medical, Victoria, Australia) designed for minimally invasive skin sampling were used for MB collection (Fig 1B). This tool provides minimally invasive skin sampling to a depth of 260 µm, corresponding to the depth of penetration of the sand fly proboscis [14], and removes a similar amount of tissue cells as the female sand fly does - the average DNA mass sampled from *M. shawi* skin (45 ng) was similar to the average DNA mass in *P. papatasi* bloodmeal (52 ng, S1 Appendix). Each MB sample was transferred into a drop of

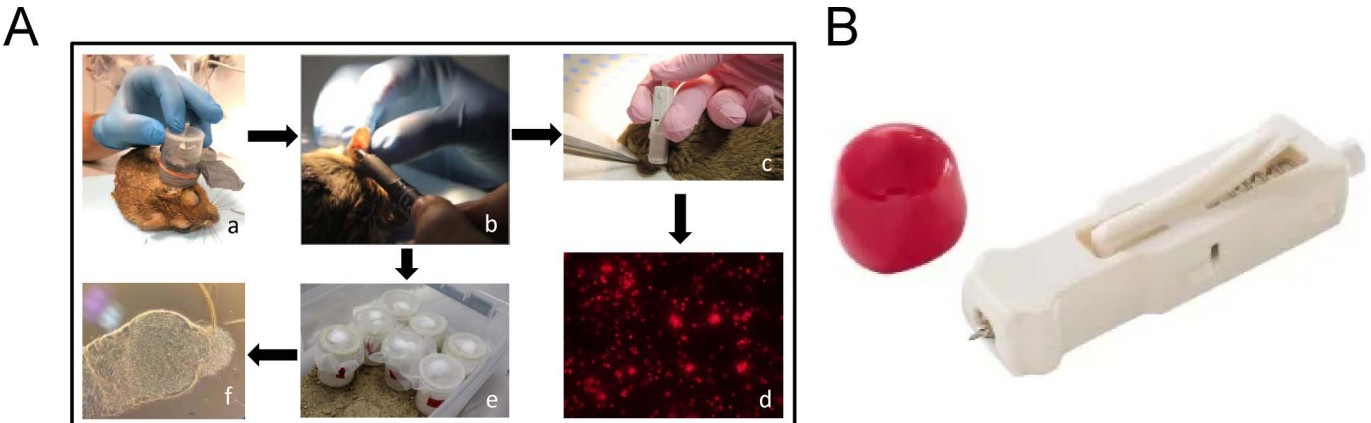

**Fig 1. Methodology of microscale examination of infectiousness.** (A) Schematic representation of the experimental procedure: a, feeding of individual *P. papatasi* females on the infected pinna; b, marking the bite site; c, taking a MB from the marked site; d, determination of the number of amastigotes by fluorescence microscopy; e, individual maintenance of engorged females for 8 days; f, examination of the sand fly gut under a light microscope. (B) Harpera skin microbiopsy tool (microsampling image courtesy of Trajan Scientific and Medical, printed with permission of the company).

saline, and the number of amastigotes was determined via fluorescence microscopy via an Olympus BX-51 microscope with an Olympus DP-72 camera and categorized as ≤ 10, 10–999, or ≥ 1000. Engored females were reared separately under standard conditions, and on day 8 PBM, their guts were examined under a light microscope for the presence of *Leishmania* infection (Fig 1A).

### Amastigote-initiated sand fly infections

Macrophage-derived amastigotes of *L. major* were obtained as described previously [15]. Mouse bone marrow precursor cells were differentiated into macrophages by cultivation in RPMI 1640 HEPES supplemented with 20% L929 fibroblast supernatant (as a source of macrophage colony stimulating factor (M-CSF), 10% FBS, 50 mM mercaptoethanol), and a mixture of antibiotics and amino acids (L-glutamine 200 mM-peniciline 10 000 U-streptomycine 10 mg/ml) (Sigma–Aldrich). The cells were incubated at 37 °C and 5% $CO_2$ for 7–10 days.

The macrophages were then infected with stationary-phase promastigotes at a ratio of 3 parasites per macrophage. After 72 hours, the macrophages were lysed in M199 (Sigma–Aldrich) medium containing 0.016% SDS for up to 7 minutes, followed by neutralization with M199 supplemented with 20% FBS. Mechanical disruption was completed by using the rubber end of a syringe plunger to scrape adherent cells off the culture plate, followed by the release of amastigotes through repeated aspiration into a 1 mL insulin syringe. The amastigotes were then washed three times by centrifugation at 3010 × g for 10 min.

Female *P. papatasi* (5–7 days old) were allowed to feed through a chick-skin membrane with a suspension of recovered amastigotes mixed 1:10 with heat-inactivated defibrinated sheep blood at four different concentrations (C1–C4): $1.4 \times 10^5$/mL, $1.4 \times 10^4$/mL, $7.1 \times 10^3$/mL and $2.8 \times 10^3$/mL. Assuming that female *P. papatasi* ingest, on average, 0.7 μl of blood through the chick-skin membrane [16], these concentrations correspond to 100, 10, 5 and 2 amastigotes per engorged female. Blood-fed females were separated immediately after feeding, kept at 25 °C with free access to 50% sugar solution and dissected for microscopic observations on day 10 PBM.

On day 10 PBM, females were dissected in drops of saline. The individual guts were examined under a light microscope, and the intensity and localization of infections were evaluated as described previously [17]. Parasite loads were graded as light (<100 parasites per gut), moderate (100–1000 parasites per gut) or heavy (>1000 parasites per gut) on the basis of [18]. The experiments were performed in duplicate.

## Extraction of DNA and *Leishmania* detection by qPCR

Extraction of total DNA from rodent tissues was performed via a DNA tissue isolation kit (Roche Diagnostics, Indianapolis, IN, USA) according to the manufacturer's instructions. Parasite quantification by quantitative PCR (qPCR) was performed in a Bio-Rad iCycler & iQ Real-Time PCR System via the SYBR Green detection method (SsoAdvanced Universal SYBR Green Supermix, Bio-Rad, Hercules, CA, USA). Primers targeting a 116 bp long kinetoplast minicircle DNA sequence (forward primer (13A): 5′-GTG GGGGAGGGGCGTTCT-3′ and reverse primer (13B): 5′-ATTTTACACCAACCC CCAGTT-3′) were used [19]. One microliter of DNA was used per individual reaction. PCR amplification was performed in triplicate under the following conditions: 3 min at 98 °C followed by 40 repetitive cycles: 10 s at 98 °C and 25 s at 61 °C. The PCR mixture was used as a negative control. A series of 10-fold dilutions of *L. major* promastigote DNA, ranging from $5 \times 10^3$ to $5 \times 10^2$ parasites per PCR, was used to prepare a standard curve. The quantitative results were expressed via interpolation with a standard curve. To monitor nonspecific products or primer dimers, a melting analysis was performed from 70 to 95 °C at the end of each run, with a slope of 0.5 °C/c and 5 s at each temperature.

## Statistical analysis

We employed [1] linear models (lm) to compare the numbers of parasites in whole ear pinnae with different disease symptoms (function lm, natural log-transformed, factor levels: asymptomatic, lesion, nodulus); [2] linear mixed-effects models (lmm: function lme as implemented in the package nlme [20] to compare numbers of parasites (natural log-transformed) in symptomatic pinna zones (factor levels: lesion centre, lesion margin, intact skin); [3] nonparametric Wilcoxon matched-pair test to compare the numbers of amastigotes in the swollen area versus intact skin of swollen ear pinnae; and [4] generalized linear model (function glm, binomial distribution, logit link) to test the effect of amastigote concentration on the proportion of females that became infected while feeding; [5] marginal generalized linear models (function geeglm, binomial distribution, logit link) as implemented in the package geepack (Generalized Estimating Equation Package; [21]) to test the effects of pinna zones and local amastigote concentration on infectivity. For the theoretical background of marginal models, see [22]. The calculations were performed in the R environment [23].

## Results

### Parasite load and spatial distribution of *L. major* in inoculated ears

The number of parasites in the entire ear pinnae was quantified by qPCR (Fig 2A and Table 1). In asymptomatic ears, the lowest parasite loads (894 thousand on average) were found. Two of these ears were from animals euthanized at 15 and 16 weeks p.i., likely presymptomatic (lesions may have developed if the experiment continued longer), and one ear was from an animal at 30 weeks p.i. with a healed nodule. The ear pinnae of symptomatic animals were typically analysed at weeks 25 and 30 p.i. An average of 1.76 million parasites were found in five swollen ears, and the highest load was present in 20 ear pinnae with ulcerative lesions (an average of 1.9 million), although the difference between the groups was not statistically significant (lm: $F_{(2,25)} = 2.66$, $P = 0.09$).

Symptomatic ears of *M. shawi* with skin lesions were further divided into three specific zones: the lesion center, lesion margin and intact skin (Fig 2B and Table 1). These zones were clearly distinguishable in 18 of the 20 animals; the remaining two were excluded from the analysis. The number of amastigotes differed sharply between the zones (lmm: $F_{(2,34)} = 31.69$, $P < 0.0001$). The lowest parasite load was detected in the intact skin (intercept = 5.659, SE = 0.961, t = 5.88, $P < 0.0001$), whereas the highest parasite load—up to tens of billions, i.e., 5 orders of magnitude greater—was detected in the centres of the lesions (coefficient = 7.898, SE = 1.068, t = 7.40, $P < 0.0001$). The difference between intact skin margins and lesion margins was also significant (coefficient = 6.669, SE = 1.067, t = 6.25, $P < 0.0001$), but the difference between lesion margins and centers was not significant (by 1–2 orders of magnitude, t = -1.15, $P = 0.2579$). Similarly, in swollen ear pinnae, significantly fewer parasites were detected in the intact skin than in the swollen area (Wilcoxon matched-paired

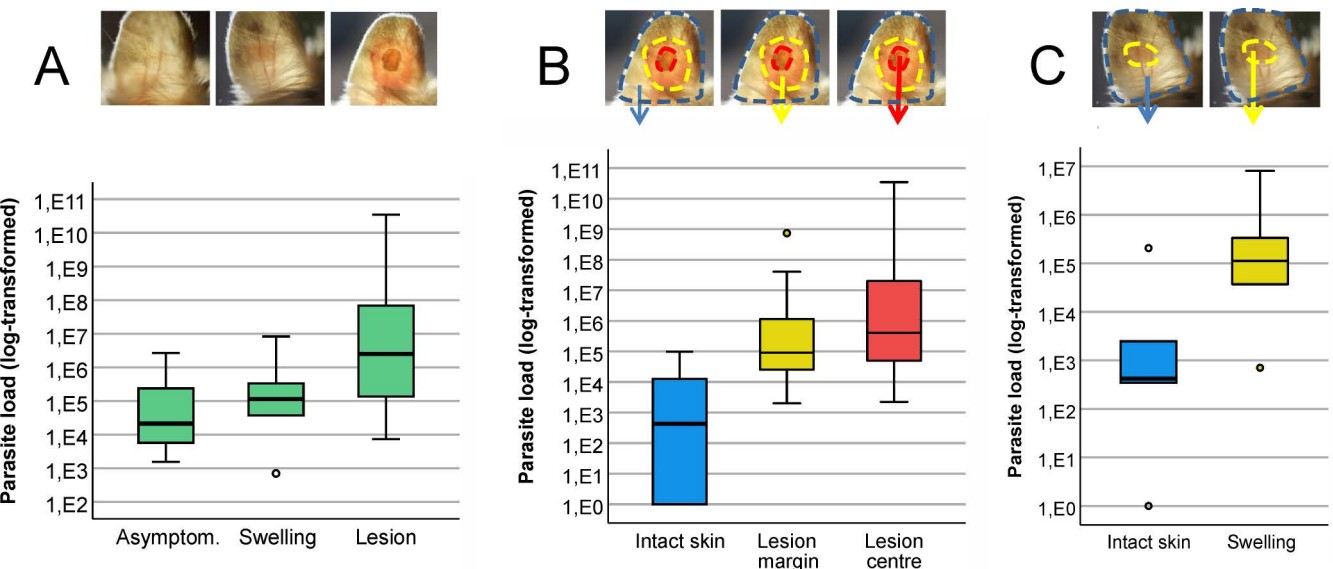

**Fig 2. *Leishmania major* loads in ear pinnae of *Meriones shawi* determined by qPCR (original data in S1 Table).** (A) Comparison of the number of parasites in asymptomatic ear pinnae, in pinnae with swelling and in pinnae with ulcerating skin lesions. (B) Parasite load in three zones of the ear pinnae with ulcerative skin lesions. (C) Parasite load in two zones of the ear pinnae with swelling. In the boxplots, the box is bordered by upper and lower quartiles (IQRs, interquartile ranges), the horizontal line denotes the median value, the whiskers denote 1.5 times the IQR, and the circles denote outliers. The images of the ear pinnae were adapted from [10].

**Table 1. Quantification of *Leishmania major* in ear pinnae of *Meriones shawi* and BALB/c mice.**

| Skin areas | | M. shawi | | | BALB/c mouse | | |
|---|---|---|---|---|---|---|---|
| | | N | Mean (thousands) | Min – Max (thousands) | N | Mean (thousands) | Min – Max (thousands) |
| Whole pinnae | Asymptomatic | 3 | 894 | 1.6 - 2 660 | – | | |
| | Swollen | 5 | 1 757 | 0.7 - 8 300 | – | | |
| | Lesion | 20 | 1.9E6 | 7.5 - 3.5E7 | – | | |
| | | | | | | | |
| Zones of pinnae with ulcerative lesion | Intact skin | 18 | 15.3 | 0 - 98 | 5 | 2.0 | 0.1 - 7.4 |
| | Lesion margin | 18 | 45 098 | 2.0 - 7E5 | 5 | 2 219 | 16.6 - 5 406 |
| | Lesion centre | 18 | 1.9E6 | 2.2 - 3E7 | 5 | 4 529 | 2 058 -6 040 |
| | | | | | | | |
| Zones of swollen pinnae | Intact skin | 5 | 41.8 | 0 - 206 | – | | |
| | Swelling | 5 | 1 715 | 0.7 - 8 090 | – | | |

The numbers were determined via qPCR. The original data are shown in S1 Table.

test, $z = 2.023$, $P = 0.0431$). The parasite load in the swellings was similar to that in the lesion margins (Fig 2C and Tables 1 and S1).

In BALB/c mice, the infection progressed uniformly, with all infected animals developing ulcerative lesions starting at week 5 p.i. Similarly, significantly fewer parasites were found in the intact areas of the skin (lmm; ANOVA: $F_{(2,8)} = 38.78$, $P < 0.0001$; model: intercept = 6.672, SE = 0.732, $t = 9.11$, $P < 0.0001$) than in the lesion centers (coefficient = 8.591, SE = 1.018, $t = 8.44$, $P < 0.0001$) and lesion margins (coefficient = 6.942, SE = 1.018, $t = 6.82$, $P < 0.0001$) (Fig 3A).

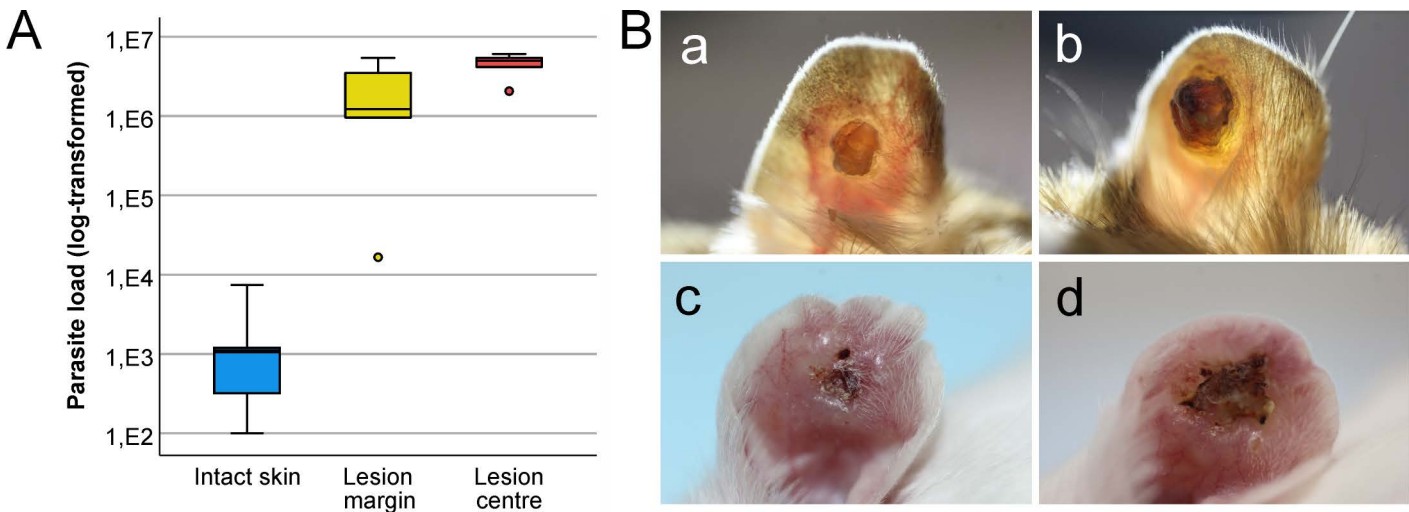

**Fig 3. *Leishmania major* loads in the ear pinnae of BALB/c mice and skin manifestations of the disease (original data in** S1 Table**).** (A) Parasite load in three zones of pinnae with ulcerative skin lesions determined by qPCR. In the boxplots, the box is bordered by upper and lower quartiles (IQRs, interquartile ranges), the horizontal line denotes the median value, the whiskers denote 1.5 times the IQR, and the circles denote outliers. (B) Appearance of skin lesions in *M. shawi* at week 25 p.i. (a, b) and BALB/c mice (c, d) at week 10 p.i. The image (a) was adapted from [10].

However, the lesions were more ulcerative than those of *M. shawi*, gradually affecting and nearly destroying the entire pinna (Fig 3B).

### Infectiousness at the microscale

Given the significant variation in the distribution of *L. major* in different zones of the ear pinnae, we next evaluated which zones were most likely sources of infection for the vectors. A total of 161 sand flies and MBs from bite sites in 24 *M. shawi* and 5 BALB/c mice were analyzed (S2 Table).

In *M. shawi*, the highest number of amastigotes was found in the MB samples taken from the centers of the lesions, with over 1000 amastigotes per sample. In contrast, the numbers of amastigotes in the lesion margins and swelling areas typically did not exceed 1000, while no amastigotes or only minimal numbers were found in the intact skin (Fig 4A). These results are consistent with those of the previous PCR analysis of the ear zones. Given the uneven distribution of *Leishmania* in pinna, it is not surprising that only 11.1% of female sand flies acquired infection (12 of 108). Surprisingly, however, no gut infections were observed under a light microscope 8 days post blood meal in *P. papatasi* females that fed at the centers of the lesions (Fig 4C). Infectiousness was not associated with the number of amastigotes in MB (df = 1, $\chi^2$ = 2.97, P = 0.085, removed from the final geeglm) but differed between the two remaining pinna zones (df = 1, $\chi^2$ = 6.79, P = 0.0092). In intact skin, only one female out of 71 (1.4%) became infected (Fig 4C). The most infectious sources were lesion margins and swellings (intercept$_{intact skin}$ = -3.265, SE = 0.997, Wald = 10.73, P = 0.0011, coefficient = 2.752, SE = 1.056, Wald = 6.79, P = 0.0092), with tens or hundreds of parasites in the MB. Here, 37. 9% of the females (11 of 29) were successfully infected.

In BALB/c mice, high numbers of amastigotes (exceeding 1,000) prevailed in MB taken in the centers of lesions as well as from the margins (Fig 4B). In total, one-third of the female sand flies that fed on the BALB/c mice became infected (10 out of 30). Specifically, 50.0% of the females that fed at the lesion centers and 58.3% of those that fed at the lesion margins acquired the infection (Fig 4D). Similar to findings with *M. shawi*, a smaller proportion of females (7.1%) were infected after feeding on intact skin; however, the effect of the pinna zone on infectiousness was not

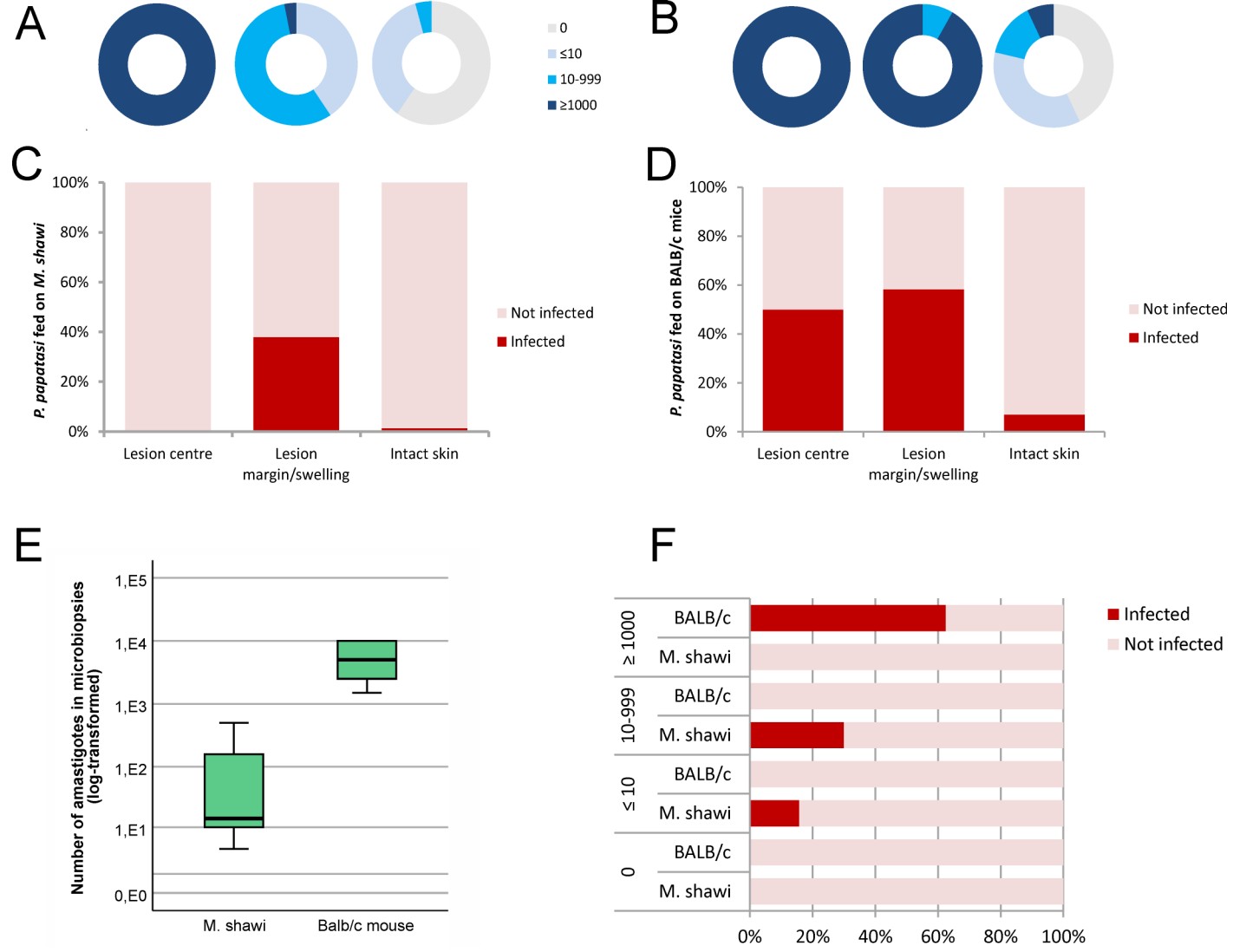

**Fig 4. Infectiousness of *M. shawi* and BALB/c mice at the microscale.** (A and B) Relative representation of the numbers of amastigotes in MB taken from infected ear pinnae with ulcerative lesions of *M. shawi* (A) and BALB/c mice (B). Left, lesion center; middle, lesion margin/swelling; right, intact skin. (C and D) Representation of infected and noninfected females of *P. papatasi* fed different zones of pinna with ulcerative lesions of *M. shawi* (C) and BALB/c mice (D). (E) Numbers of amastigotes in MB taken from the pinna of infected *M. shawi* and BALB/c mice. In the boxplots, the box is bordered by upper and lower quartiles (IQRs, interquartile ranges), the horizontal line denotes the median value, the whiskers denote 1.5 times the IQR, and the circles denote outliers. (F) Representation of infected and noninfected females of *P. papatasi* fed at sites with different numbers of amastigotes (revealed by MB) on pinna with ulcerative lesions of *M. shawi* and BALB/c mice. The source data are available in S2 Table.

statistically significant (df = 2, χ2 = 4.117, P = 0.1276). In the final Gomeglm model, the only significant predictor (df = 1, χ2 = 15.2, P < 0.001) was the number of amastigotes in the MB (intercept = -4.118, SE = 0.791, Wald = 27.1, coefficient = 0.525, SE = 0.135, Wald = 15.2, P < 0.0001). Females that fed in areas where no amastigotes were detected in MB were never infected. In *M. shawi,* the number of amastigotes collected by MB at bite sites where females became infected ranged from 4-500 (Fig 4E). Sand fly infections were successful in 16% of the cases (6 out of 38) at sites where MB revealed fewer than 10 amastigotes and in 30% of the cases (6 of 20) at sites where 10–999 amastigotes were present in the MB (Fig 4F). On the other hand, none of the *P. papatasi* females that fed on BALB/c mice became

infected at sites with fewer than 1000 parasites in MB. However, 62% (10 of 16) of the females that fed at sites with more than 1000 parasites in the MB became infected, and the number of amastigotes at sites of successful infection ranged from 1,500–10,000 (Fig 4E and 4F).

Since the MB collection technique is noninvasive, it was possible to test the same animals repeatedly. Thus, samples from the same pinna of 13 *M. shawi* were evaluated at both 15 and 25 weeks p.i. During this 10-week period, asymptomatic animals developed skin lesions, and the number of amastigotes in the MB generally increased (Table 2). By week 25 p.i., half of these animals (4 from 8) were also infectious to the sand flies. Among the three animals that were already symptomatic at week 15, two also presented an increase in the number of AMs in the MB by week 25. In three individuals where the number of amastigotes in MB did not increase, sand flies fed (and thus MB were taken) predominantly on the intact areas of the ear pinnae. Four BALB/c mice were also evaluated 2–3 weeks apart (between weeks 7–9 and weeks 10–11), but no significant difference was observed, as their pinnae already contained high numbers of amastigotes in the first sampling period, and the mice were already infectious to vectors.

### Amastigote-initiated infections of *P. papatasi*

To confirm that a low infectious dose of 1–10 amastigotes is sufficient to colonize the gut of a female sand fly, we performed a series of experiments using 4 different concentrations of amastigotes: C1, C2, C3 and C4, corresponding to 100, 10, 5 and 1 amastigote per 0.7 µl, respectively. This volume represents the average amount ingested by a single female *P. papatasi* via the chick-skin membrane [16]. The infection rate was affected by the treatment (glm: df = 3,132, Deviance 30.66, Residual deviance = 115.34, $\chi2 = 33.355$, P < 0.0001): the proportion of infected females at the C1 concentration (intercept = 0.272, SE = 0.332, z = 0.819, P = 0.4125) was significantly greater than the proportion at the C2–C4 concentration (C2: coefficient = -2.351, SE = 0.626, z = -3.759, P = 0.0002; C3: coefficient = -2.575, SE = 0.690, z = -3.729, P = 0.0002; C4: coefficient = -2.469, SE = 0.6931, z = -3.562, P = 0.0004). However, the differences between groups C2--C4 were not statistically significant. In these *Leishmania*-positive females in groups C2–C4, light or moderate infections predominated (Fig 5), yet the stomodeal valve was consistently colonized, which is a crucial step for successful transmission of the parasite to the next host.

### Discussion

This study offers novel insights into the transmission of *L. major*, emphasizing the microscale spatial distribution of parasites in the natural host, *M. shawi,* and their associated infectiousness to insect vectors.

Our previous experimental study on *M. shawi* demonstrated that at the macroscale, the highest load of *L. major* persisted for several months in infected ear pinnae, whereas smaller parasite numbers spread to ear-draining lymph nodes and other skin areas, the spleen and liver [10]. Field studies have also shown that the ear pinnae are naturally the preferred bite sites for sand flies, with lesions predominantly localized on this body part in *M. shawi* [24]. Hence, the ear pinnae were the focus of the detailed microscale analysis presented here. Despite a low infectious dose, the vast majority (93%) of *M. shawi* developed lesions that persisted for several weeks. As a result, the availability of material from asymptomatic ear pinnae was limited, comprising only two presymptomatic and one healed pinna, all of which exhibited a

**Table 2. Trend in parasite load between weeks 15 and 25 p.i. in MBs collected from the same *M. shawi* ear pinnae.**

| External manifestation week 15 p.i.>>>> week 25 p.i. | Number of amastigotes in MBs | No. of animals |
|---|---|---|
| Asymptomatic>>>> Skin lesion | Increased | 8 |
| | Stable | 2 |
| Skin lesion>>>> Skin lesion | Increased | 2 |
| | Stable | 1 |

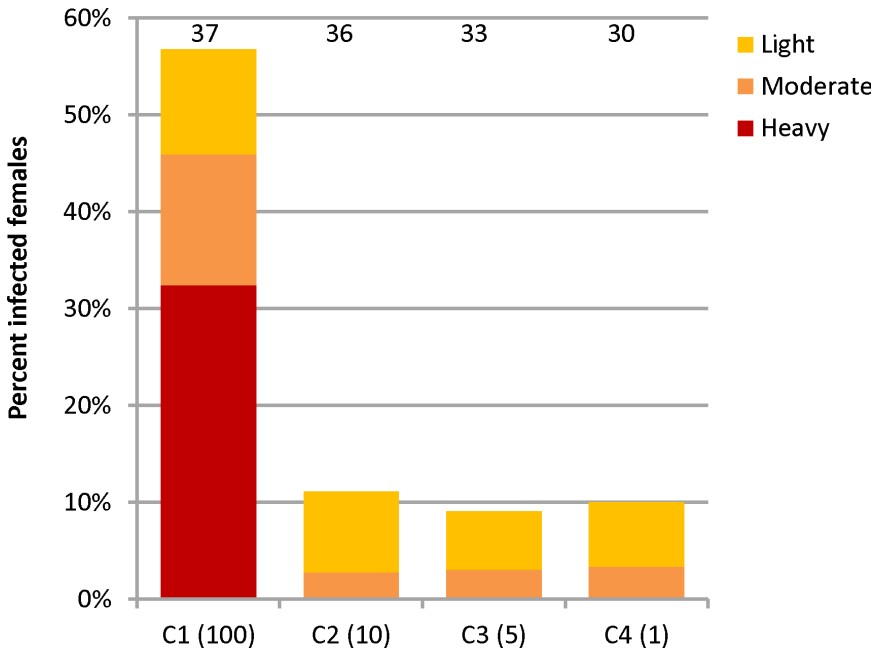

**Fig 5. Infection rates of *P. papatasi* females fed different infective doses.** The concentrations correspond to 100 (C1), 10 (C2), 5 (C3) or 1 (C4) amastigotes per female. The numbers above the columns indicate the number of dissected females.

lower parasite load compared to symptomatic animals. However, the difference in parasite numbers was not statistically significant, which aligns with our previous study demonstrating that the presymptomatic *M. shawi* and animals with healed skin symptoms can still serve as infectious hosts for vector [10].

Xenodiagnostic experiments have revealed that even in skin areas with a high parasite load, only a subset of sand fly females becomes infected. Typically, only 10–30% develop infection, a pattern consistently reported across various vector–host–parasite combinations [25–27]. In our study, we observed a similarly low infection rate (11%) in females that fed on ear pinnae of *M. shawi* with skin lesions. These results suggest a patchy distribution of parasites in the host skin, similar to the pattern observed for *L. donovani* in RAG mice [9]. While the skin of mice infected with *L. donovani* is asymptomatic [9,26] and the patchiness of *L. donovani* in mice appears random [9], *L. major* typically induces skin lesions at the bite site. Thus, the infected ear pinna is naturally divided into a symptomatic part (with a distinct lesion centers and margins) and surrounding apparently unaffected skin. Our results revealed that the *L. major* burden was significantly higher in symptomatic skin areas than in intact areas of the ear pinnae. Interestingly, parasite numbers in the lesion margins were similar to those found in the nodules and swellings, which represent earlier stages preceding the formation of the ulcerating lesion.

As expected, the infectiousness of different parts of the ear pinnae varied, but surprisingly, we observed a substantial difference between the two models. In *M. shawi*, sand fly females that fed on lesion margins, but not those that fed in lesion centers, became infected. In contrast, in BALB/c mice, there was little difference in infectiousness between the lesion margins and centers. We suggest that this discrepancy may stem from differences in the skin environment between these two hosts. In BALB/mice, with less effective immune defense, lesions appeared within 4 weeks, spread quickly, and at approximately 10 weeks, affected the entire pinna. In *M. shawi,* lesions appeared much later, between 9 and 15 weeks p.i., and often healed over several months [10]. The more efficient immune processes involved in skin healing in this natural host were reflected by the dry appearance of the lesion centres (see Fig 3), which may have negatively affected the vitality of amastigotes at this site, thereby preventing infection of the vector. In immunocompetent humans, who can

also self-cure *L. major* infection within a few months, the immune response leading to self-healing is associated with a Th1-mediated IFNγ response [28]. Recent studies in humans [29] infected with *L. major* naturally acquired through sand fly bites have highlighted that immune responses differ between the lesion core and ulcer (lesion center), with distinct immune niches characterized by selective chemokine/cytokine expression. These responses are linked to the cellular composition and various stages of epidermal remodelling associated with ulceration or underlying inflammation in affected human skin [29]. A more detailed investigation of these interactions between natural hosts and parasites warrants further in-depth research.

Rodents in nature are in constant contact with sand flies, which frequently nest in their burrows [6]. Repeated exposure to multiple bites from uninfected sand flies affects the progression of infection. Vojtkova et al. [27] reported that *L. major*-infected BALB/c mice exposed to naïve sand flies at two-week intervals developed skin lesions more rapidly and exhibited a higher parasite load. In our experiment, animals were exposed to sand flies at five-week intervals during xenodiagnosis. Thus, under natural conditions, more frequent exposure to sand fly saliva may accelerate lesion development and increase parasite burden at affected sites. However, this hypothesis requires experimental validation, ideally using *M. shawi* or another natural host model.

Another interesting finding was that a very small number of amastigotes (10 or fewer) in the margins of *M. shawi* lesions were sufficient to infect the vector. The minimum infection load is a key parameter for modelling transmission dynamics [9], yet experimental data on this threshold are limited. It has been reported that the presence of 1–2 promastigotes is sufficient to experimentally infect sand flies through a chick skin membrane [16,30]. However, a higher threshold was expected for amastigote-initiated infections, since there is a significant decrease in the number of *Leishmania* transforming from the amastigote to the promastigote form in the sand fly digestive tract during the first 24 h, before promastigotes begin to multiply [31]. To determine the minimum infection load in our model, we used special MB devices [14] to simulate the bite of a female sand fly. This method showed that sand flies feeding on *M. shawi* could successfully become infected even at sites where fewer than 10 amastigotes were present. This surprising result was further confirmed by a series of sand fly infections through the chick-skin membrane using different infective doses. Successful infections were observed in 8–10% of *P. papatasi* females infected with doses corresponding to 1–10 amastigotes per bloodmeal. In a previous study, Anjili et al. (2006) [32] reported that blood containing a single amastigote of *L. major* per 0.5 µl caused infections in 8.1% of membrane-fed *P. duboscqi*, although they did not report the intensity and localization of infections in the sand fly gut. In our experiment, all females infected with low infective doses (1–10 amastigotes per blood meal) developed low- or moderate-intensity infections by day 8 PBM (with fewer than 100 or 1000 promastigotes in the gut). However, their stomodeal valves were always colonized with haptomonads, which are crucial for successful transmission to the next host [33]. Moreover, the number of promastigotes in the gut may increase with subsequent blood meals [34,35], meaning that even females with weaker infections could still contribute to parasite transmission during their lifetime.

However, transmission parameters in the BALB/c mouse model significantly differed. If this study relied solely on this animal model, we would have reported a much higher minimum infection threshold, as female *P. papatasi* fed at sites where fewer than 1000 amastigotes were present did not become infected. The threshold observed in BALB/c mice aligns with those (500–1000) calculated by Doehl et al. (2017) [9] in an immunodeficient mouse model infected with *L. donovani*.

In summary, the distribution of *L. major* in *M. shawi* ear pinnae is not uniform, with the vector being most effectively infected at the lesion margins, where even a very small number of ingested amastigotes can initiate vector infection. The results obtained from the laboratory immunodeficient model should be interpreted with caution, as parasite–host interactions may differ qualitatively and quantitatively in natural hosts with a long coevolutionary history with the parasite.

## Supporting information

**S1 Appendix. DNA mass from the blood meals of *Phlebotomus papatasi* and from skin microbiopsies.**
(DOCX)

**S1 Table. Numbers of amastigotes in different zones of ear pinnae of *M. shawi* inoculated with *L. major.***
(XLSX)

**S2 Table. Numbers of amastigotes in microbiopsies taken at bite sites of *Phlebotomus papatasi* on *Meriones shawi.***
(XLSX)

## Acknowledgments

We sincerely thank Eva Gluenz and Tom Beneke for their help with the fluorescent labelling of *Leishmania* and Alon Warburg for training in the use of microbiopsies. We also thank Lenka Krejčiříková, Lenka Hlubinková and Kristýna Srstková for their technical and administrative assistance and Barbora Matějková for taking care of the colony of *Phlebotomus papatasi*.

## Author contributions

**Conceptualization:** Jovana Sádlová.

**Formal analysis:** Daniel Frynta.

**Funding acquisition:** Petr Volf.

**Investigation:** Barbora Vojtková, Tomáš Bečvář, Lenka Pacáková, Jovana Sádlová.

**Methodology:** Barbora Vojtková.

**Resources:** Nalia Mekarnia, Kamal Eddine Benallal.

**Supervision:** Petr Volf, Jovana Sádlová.

**Writing – original draft:** Jovana Sádlová.

**Writing – review & editing:** Petr Volf.

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
