## [Decision Letter · Decision Letter 0]

Infectiousness of Leishmania major to Phlebotomus papatasi: differences between natural reservoir host Meriones shawi and laboratory model BALB/c mice.

Dear Dr. Sadlova,

Thank you for submitting your manuscript to PLOS Neglected Tropical Diseases. After careful consideration, we feel that it has merit but does not fully meet PLOS Neglected Tropical Diseases's publication criteria as it currently stands. Therefore, we invite you to submit a revised version of the manuscript that addresses the points raised during the review process.

Please submit your revised manuscript within 60 days. If you will need more time than this to complete your revisions, please reply to this message or contact the journal office at plosntds@plos.org. Please include the following items when submitting your revised manuscript:

We look forward to receiving your revised manuscript.

Kind regards,

Juliana Menezes

Academic Editor

Laura-Isobel McCall

Section Editor

Shaden Kamhawi

co-Editor-in-Chief

Paul Brindley

co-Editor-in-Chief

**Journal Requirements:**

1) We do not publish any copyright or trademark symbols that usually accompany proprietary names, eg ©,  ®, or TM  (e.g. next to drug or reagent names). Therefore please remove all instances of trademark/copyright symbols throughout the text, including:

- ® on page: 10

- TM on page: 10.

3) Some material included in your submission may be copyrighted. According to PLOSu2019s copyright policy, authors who use figures or other material (e.g., graphics, clipart, maps) from another author or copyright holder must demonstrate or obtain permission to publish this material under the Creative Commons Attribution 4.0 International (CC BY 4.0) License used by PLOS journals. Please closely review the details of PLOSu2019s copyright requirements here: PLOS Licenses and Copyright. If you need to request permissions from a copyright holder, you may use PLOS's Copyright Content Permission form.

Potential Copyright Issues:

i) Please confirm (a) that you are the photographer of 1, 2, 3A, and 3B, or (b) provide written permission from the photographer to publish the photo(s) under our CC BY 4.0 license.

4) Please amend your detailed Financial Disclosure statement. This is published with the article. It must therefore be completed in full sentences and contain the exact wording you wish to be published.

1) State the initials, alongside each funding source, of each author to receive each grant. For example: "This work was supported by the National Institutes of Health (####### to AM; ###### to CJ) and the National Science Foundation (###### to AM).".

**Comments to the Authors:**

**Please note that two reviews are uploaded as attachments.**

**Reviewers' Comments:**

Reviewer's Responses to Questions

**Key Review Criteria Required for Acceptance?**

**Methods**

-Are the objectives of the study clearly articulated with a clear testable hypothesis stated?

-Is the study design appropriate to address the stated objectives?

-Is the population clearly described and appropriate for the hypothesis being tested?

-Is the sample size sufficient to ensure adequate power to address the hypothesis being tested?

-Were correct statistical analysis used to support conclusions?

-Are there concerns about ethical or regulatory requirements being met?

Reviewer #1: The manuscript PNTD-D-25-00050 entitled "Infectiousness of Leishmania major to Phlebotomus papatasi: differences between natural reservoir host Meriones shawi and laboratory model BALB/c mice." is a well designed study on the transmission parameters of L. major in two rodent models. Authors provide important aspects of the infection on the natural reservoir host of this parasite, comparing the obtained data with laboratory model rodents. The manuscript is well written, and deserves publication in PNTD pending minor revisions.

Bellow I provided some comments and suggestions to improve the quality of the manuscript.

Abstract:

-Use "ear pinnae" intead of "pinnae".

Introduction:

-Second paragraph: Please add some information on the vector of L. major. Also, replace the sentence "These reservoir animals are the main source of infection for vectors, phlebotomine sand flies (Diptera: Psychodidae), which act as vectors, transmitting parasites to humans, who are incidental hosts" by "These reservoir animals are the main source of infection for phlebotomine sand flies (Diptera: Psychodidae), which act as vectors, transmitting parasites to other vertebrate hosts, including humans, who are incidental hosts".

-Second paragraph: Add "For example," before "Heterogeneity".

-Last paragraph: The sentences "Amastigote numbers at bite sites were estimated via microbiopsies (MBs), which sampled a volume comparable to what the feeding female sand fly takes by her proboscis. The results from the natural host model were compared with those from a standard laboratory model using BALB/c mice." are methods and should be deleted from the introduction.

Methods:

-Please use the term "euthanasia" instead of "sacrifice".

Reviewer #2: The study's objectives should be presented clearly and concisely.

The authors employed various statistical models to interpret the results; however, they did not specify the number of sandflies used in the infection experiments or the number of natural hosts and laboratory animals included in the study following the sample size calculation procedure.

Reviewer #3: Based on the overall quality of the study, the relevance of the topic, and the robustness of the data presented, the manuscript is suitable for publication pending minor revisions. The suggested clarifications and additional analyses will further strengthen the manuscript, enhancing the understanding of parasite distribution and infectiousness in natural and experimental models.

**Results**

-Does the analysis presented match the analysis plan?

-Are the results clearly and completely presented?

-Are the figures (Tables, Images) of sufficient quality for clarity?

Reviewer #1: Results:

-This section could be better summarized as there are a lot of information that fits on material and methods. Please try to keep only the analytical part on the results section.

- Figure 3A and 3B are part of methodology and should be moved to the M&M section along with the text to which it was referred.

Reviewer #2: The first paragraph of the results section confuses methods and results.Figure 4 is confusing because the authors have used different notations, such as 2x and 1x, without clear explanation.

Reviewer #3: The results are generally well presented, but some methodological details, such as the rationale for infection time points and the low sample size of asymptomatic animals, require further explanation.

The figures are informative, but minor corrections are needed, including improving Figure 1 arrows, standardizing data representation in Figure 3, adding the missing Y-axis value in Figure 5, and replacing the schematic graph in Figure 4 with a numerical table.

**Conclusions**

-Are the conclusions supported by the data presented?

-Are the limitations of analysis clearly described?

-Do the authors discuss how these data can be helpful to advance our understanding of the topic under study?

-Is public health relevance addressed?

Reviewer #1: -The first paragraph of discussion should be mover to the end of introduction as it states justification and aim of the study. Please start the discussion by focusing on the main findings.

-Please rephrase the sentence "The centers of the lesions remained dry (see also Fig 2), and the more efficient immune processes involved in skin healing may have negatively affected the vitality of amastigotes at this site, preventing them from infecting the vector.” to make it clearer.

Reviewer #2: The conclusion would be much more impactful if articulated more clearly and directly.

Reviewer #3: the study highlights the importance of using natural hosts to improve our understanding of parasite transmission dynamics, however the limitations regarding sample size and infection time points are not explicitly addressed and should be better discussed.

**Editorial and Data Presentation Modifications?**

Reviewer #1: (No Response)

Reviewer #2: (No Response)

Reviewer #3: Minor revisions

The text is concise and clearly describes the objectives and results of the project.

1- Introduction:

Line 16- The sentence should be revised: These reservoir animals are the main source of infection for vectors, phlebotomine sand flies (Diptera: Psychodidae), which act as vectors, transmitting parasites to humans, who are incidental hosts.

Please expand the introduction. Specifically, include information on how P. papatasi serves as a specific vector for L. major, highlighting key aspects of factors influencing vector competence.

2- Methods and Results

Experimental infections of rodents and Infectiousness of rodents at the microscale: (The experiment was terminated at 15–38 p.i. in Meriones shawi - )

It is unclear what infection time points were analyzed for M. shawi, and the reason for the long time gap between the selected post-infection intervals is not explained. Could you clarify the rationale behind these choices?

The number of infected and asymptomatic animals analyzed appears to be low, with only three asymptomatic ears examined. Additionally, how does this sample size impact the robustness of the conclusions regarding parasite load dynamics in M. shawi?

Fig 1 – The arrow in Figure 1 is not appearing. Please check and adjust the figure to ensure the arrow is clearly visible.

In Figure 3, panel (E) appears to express the data as percentages, whereas panel (F) seems to use relative numbers. To ensure consistency and clarity, you should express all data in percentages.

Specifically, 50.0% of the females that fed at the lesion centers and 58.3% of those that fed at the lesion margins acquired the infection (Fig 3F). The 50% infection rate is based on only four animals. Could the small number of insects used in this analysis influence the reliability of the results?

Fig 4. Schematic of the trend of parasite loads in MBs collected from M. shawi pinnae during the experiment. It would be more valuable to present a table with the MB values found rather than a schematic graph.

Fig 5. Infection rates of P. papatasi females fed different infective doses. Please add the missing 10% value on the Y-axis

**Summary and General Comments**

Reviewer #1: (No Response)

Reviewer #2: The authors highlighted the differences in infectiousness to sandflies between natural reservoir hosts and animal models. They found that using animal models to estimate the infectiousness of parasites to vectors tends to underestimate parasite transmission in leishmaniasis. They reported that only a few parasites from natural animal models could establish an infection in sandflies, in contrast to the much higher numbers observed in animal models needed to establish infection. These findings could enhance our understanding of Leishmania transmission.

Reviewer #3: In this study, the authors show an interesting difference in the distribution of Leishmania major amastigotes in the skin of Meriones shawi and BALB/c mice revealed differences in infectiousness to sand flies. These findings underscore the importance of natural host models for a better understanding of CL transmission and more accurate epidemiological modeling. So, the current study is on a topic of relevance and general interest to the readers of the journal presenting originality and importance of the contribution for the development of the field of study.

The authors of this article have extensive experience in Leishmania-sand fly interactions and have provided valuable insights into the transmission dynamics of Leishmania major, particularly the role of Meriones shawi as a reservoir host.

I think that a histological or microscopic examination of the three zones of ulcerative lesions in both BALB/c mice and M. shawi at the same time points could provide valuable insights into the spatial dynamics of parasite distribution and tissue response.

PLOS authors have the option to publish the peer review history of their article (what does this mean? ). If published, this will include your full peer review and any attached files.

**Do you want your identity to be public for this peer review?** For information about this choice, including consent withdrawal, please see our Privacy Policy .

Reviewer #1: No

Reviewer #2: No

Reviewer #3: No

**Figure resubmission:**

**Reproducibility:**



---

## [Decision Letter · Decision Letter 1]

Dear Dr. Sadlova,

We are pleased to inform you that your manuscript 'Infectiousness of Leishmania major to Phlebotomus papatasi: differences between natural reservoir host Meriones shawi and laboratory model BALB/c mice.' has been provisionally accepted for publication in PLOS Neglected Tropical Diseases.

Best regards,

Laura-Isobel McCall

Section Editor

Laura-Isobel McCall

Section Editor

Shaden Kamhawi

co-Editor-in-Chief

Paul Brindley

co-Editor-in-Chief

Reviewer's Responses to Questions

**Key Review Criteria Required for Acceptance?**

**Methods**

-Are the objectives of the study clearly articulated with a clear testable hypothesis stated?

-Is the study design appropriate to address the stated objectives?

-Is the population clearly described and appropriate for the hypothesis being tested?

-Is the sample size sufficient to ensure adequate power to address the hypothesis being tested?

-Were correct statistical analysis used to support conclusions?

-Are there concerns about ethical or regulatory requirements being met?

Reviewer #1: Authors have correctly addressed previous comments and suggestions

Reviewer #2: (No Response)

Reviewer #3: (No Response)

**Results**

-Does the analysis presented match the analysis plan?

-Are the results clearly and completely presented?

-Are the figures (Tables, Images) of sufficient quality for clarity?

Reviewer #1: (No Response)

Reviewer #2: (No Response)

Reviewer #3: (No Response)

**Conclusions**

-Are the conclusions supported by the data presented?

-Are the limitations of analysis clearly described?

-Do the authors discuss how these data can be helpful to advance our understanding of the topic under study?

-Is public health relevance addressed?

Reviewer #1: (No Response)

Reviewer #2: (No Response)

Reviewer #3: (No Response)

**Editorial and Data Presentation Modifications?**

Reviewer #1: (No Response)

Reviewer #2: (No Response)

Reviewer #3: (No Response)

**Summary and General Comments**

Reviewer #1: (No Response)

Reviewer #2: (No Response)

Reviewer #3: After carefully reviewing the revised version of the manuscript titled "Infectiousness of Leishmania major to Phlebotomus papatasi: differences between natural reservoir host Meriones shawi and laboratory model BALB/c mice", I confirm that all previously requested modifications have been adequately addressed. The authors have improved the clarity of the text, provided the necessary methodological clarifications, and incorporated the suggested revisions that strengthen the overall interpretation of the results.

I recommend the acceptance of the manuscript for publication, with no further revisions required.

PLOS authors have the option to publish the peer review history of their article (what does this mean? ). If published, this will include your full peer review and any attached files.

**Do you want your identity to be public for this peer review?** For information about this choice, including consent withdrawal, please see our Privacy Policy .

Reviewer #1: No

Reviewer #2: No

Reviewer #3: No

---

## [Editor Report · Acceptance letter]

Dear Dr. Sádlová,

We are delighted to inform you that your manuscript, "Infectiousness of Leishmania major to Phlebotomus papatasi: differences between natural reservoir host Meriones shawi and laboratory model BALB/c mice.," has been formally accepted for publication in PLOS Neglected Tropical Diseases.

Best regards,

Shaden Kamhawi

co-Editor-in-Chief

Paul Brindley

co-Editor-in-Chief
